# Quantitative three-dimensional local order analysis of nanomaterials through electron diffraction

Ella Mara Schmidt [1,2,3] ✉, Paul Benjamin Klar [1,4], Yaşar Krysiak [4,5], Petr Svora [4], Andrew L. Goodwin [3] & Lukas Palatinus [4]

Structure-property relationships in ordered materials have long been a core principle in materials design. However, the introduction of disorder into materials provides structural flexibility and thus access to material properties that are not attainable in conventional, ordered materials. To understand disorder-property relationships, the disorder – i.e., the local ordering principles – must be quantified. Local order can be probed experimentally by diffuse scattering. The analysis is notoriously difficult, especially if only powder samples are available. Here, we combine the advantages of three-dimensional electron diffraction – a method that allows single crystal diffraction measurements on sub-micron sized crystals – and three-dimensional difference pair distribution function analysis (3D-ΔPDF) to address this problem. In this work, we compare the 3D-ΔPDF from electron diffraction data with those obtained from neutron and x-ray experiments of yttria-stabilized zirconia ($Zr_{0.82}Y_{0.18}O_{1.91}$) and demonstrate the reliability of the proposed approach.

Functional materials design uses structure–property relationships which focus on structurally ordered systems, where disorder and defects are generally considered detrimental. However, it is known that certain types of correlated disorder can lead to phenomena that are inaccessible to ordered structures[1–4]. Examples include the compositional complexity that drives domain structure in relaxor ferroelectrics (such as $PbMg_{1/3}Nb_{2/3}O_3$ or $Sr_{0.5}Ba_{0.5}Nb_2O_6$)[5,6] and relaxor ferromagnets (such as $LaNi_{2/3}Sb_{1/3}O_3$)[7] and Jahn-Teller distortions that give rise to specific electronic and magnetic properties as e.g., observed in $LaMnO_3$[8] or $PbTe$[9].

To successfully engineer local order in novel functional materials, the quantitative characterisation of the local order is essential[1,10,11]. In a diffraction experiment, long-range order manifests itself in Bragg reflections, the analysis of which is well established – sometimes even automated – in conventional crystallography. Information about local order is encoded in the much weaker diffuse scattering, the analysis of which is notoriously difficult and far from routine. Increasingly,

powder pair distribution function analysis (powder PDF) from x-ray, neutron and/or electron scattering experiments has become the most applied method for accessing local ordering motifs[10,12–16] – especially in nanocrystalline samples that are not suitable for single crystal x-ray or neutron diffraction experiments. The major drawback of the powder PDF is the one-dimensional nature of the approach: the powder PDF is a projection of three-dimensional structural information and essentially provides a histogram of interatomic distances. This may lead to ambiguities if (1) interatomic-vectors in different directions have similar lengths or (2) if several (different) disordered phases are present, as the inseparable ensemble average of all crystallites in the powdered samples is taken.

Three-dimensional single-crystal diffuse scattering can be a remedy to resolve these ambiguities, as this approach (1) preserves the structural information in 3D space and (2) is performed on single crystals. There are many different approaches to the analysis of single crystal diffuse scattering[17–19] of which the recently developed analysis

[1]Faculty of Geosciences and MAPEX Center for Materials and Processes, University of Bremen, Bremen, Germany. [2]MARUM Center for Marine Environmental Sciences, University of Bremen, Bremen, Germany. [3]Inorganic Chemistry Laboratory, University of Oxford, Oxford, United Kingdom. [4]Institute of Physics of the Czech Academy of Sciences, Prague, Czechia. [5]Institute of Inorganic Chemistry, Leibniz University Hannover, Hannover, Germany. ✉e-mail: ella.schmidt@uni-bremen.de

of three-dimensional difference pair distribution function (3D-ΔPDF) provides the most intuitive and direct interpretation[20–22]. As the name suggests, the 3D-ΔPDF quantifies difference pair correlations. It is the Fourier transform of the diffuse scattering intensity without the Bragg scattering intensities. The 3D-ΔPDF densities therefore correspond to local deviations away from the average structure, where positive 3D-ΔPDF densities correspond to interatomic vectors with more scattering density in the real structure than suggested by the average structure, while negative 3D-ΔPDF densities correspond to interatomic vectors with less scattering density than suggested by the average structure. An intuitive example would be the correlated thermal motion of neighbouring atoms: if neighbouring atoms vibrate in-phase, their interatomic vector is more confined than the average interatomic vector. In the average structure, the smeared out scattering density of each individual atom is modelled by the atomic displacement parameters by assuming uncorrelated thermal motion. The resulting signature in the 3D-ΔPDF is a sharp positive peak surrounded by a region of negative density[20].

To date, 3D-ΔPDF approaches have been successfully applied to single crystal x-ray and neutron diffraction measurements on various material classes[22–27]. However, these methods in general require micrometric single crystals (Ø > 5 μm), a requirement that is not always met for functional and applied materials, especially when properties of interest depend on the crystal size. In these cases, the analysis of larger samples from an adapted synthesis is not necessarily representative for the as-applied nano-sized material. Recent developments in the field of electron diffraction allow this limitation to be overcome: 3D electron diffraction (3D ED) experiments are routinely performed on submicron crystallites[28,29].

Historically, single crystal electron diffuse scattering from submicron crystallites has been analysed using oriented zone axis patterns[30–33], which only provide information in selected projections of the three-dimensional structure. Data acquisition using 3D ED methods can overcome this limitation[34–36] and we come to show how the resulting 3D data sets can be exploited in 3D-ΔPDF analysis, which requires full reciprocal space coverage.

The sample material that we use in the work presented here is yttria-stabilised zirconia ($Zr_{0.82}Y_{0.18}O_{1.91}$, YSZ) – a well-known and technologically important material. YSZ shows pronounced diffuse scattering for which a consistent local order model has been established in the literature[23,37–40]. Pure $ZrO_2$ adopts the cubic fluorite structure at elevated temperatures but is monoclinic at ambient conditions[38]. The cubic phase is stabilised at ambient conditions by the introduction of aliovalent oxides, such as $Y_2O_3$. The resulting compound adopts a disordered fluorite structure with a composition-dependent concentration of vacant oxygen sites as described by the general formula $Zr_{1-x}Y_xO_{2-x/2}$. The presence of oxygen vacancies is of great technological importance, as they are prerequisite for oxygen ion conduction[41]. From a structural point of view, oxygen vacancies lead to local distortions[38] as illustrated in Fig. 1: oxygen ions neighbouring a vacancy shift along the ⟨100⟩ directions towards the vacancy, whereas metal ions neighbouring a vacancy shift along the ⟨111⟩ directions away from the vacancy[23,38,40]. Furthermore, in YSZ there is a tendency to form 6-fold coordinated metal ions by vacancy pairs separated by ⟨$\frac{1}{2}\frac{1}{2}\frac{1}{2}$⟩ vectors[23,38,40]. In YSZ the static displacements due to the introduction of the vacancies are the dominant reason for the diffuse scattering, while chemical short-range order seems to be less pronounced and hence only contributes to a minor degree to the observed diffuse scattering[38].

This local order in YSZ leads to distinct signatures in the 3D-ΔPDFs measured using x-ray and neutron diffraction, which can be interpreted quantitatively in terms of a local order model[23].

In this article, we demonstrate how the 3D-ΔPDF technique can be applied to single crystal diffuse scattering data obtained from 3D ED data. YSZ is an ideal reference material to establish and test the

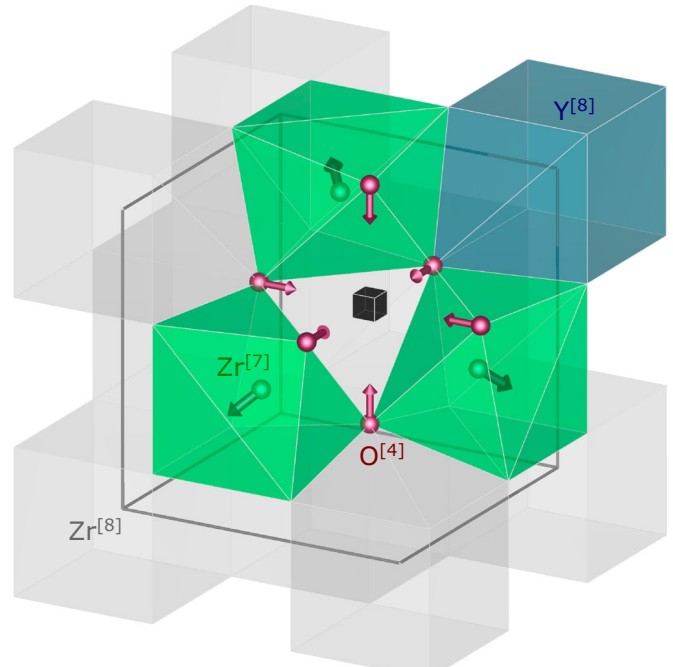

**Fig. 1 | Schematic defect model for cubic stabilised zirconia.** Grey polyhedra indicate $Zr^{4+}$ ions in regular 8-fold coordination. The solid black cube indicates a vacancy site. $Zr^{4+}$ ions in 7-fold coordination are displaced away from the vacancy along the ⟨111⟩ directions, indicated in green. $O^{2−}$ ions directly next to the vacancy are displaced towards the vacancy along the ⟨100⟩ directions, indicated in purple. A possible $Y^{3+}$ site that is the next-nearest neighbour to the vacancy is indicated by the blue coordination polyhedron. Figure generated using Vesta[62].

reliability of extracting and interpreting the electron 3D-ΔPDF: For YSZ large neutron-size single crystals are readily available and by comparing the results of the 3D-ΔPDF from x-ray and neutron experiments with our analysis of the electron diffraction data, we demonstrate the applicability and reliability of the proposed method.

## Results

### Reciprocal space

An YSZ single crystalline thin lamella was prepared by ion milling and electron diffraction patterns were measured with continuous-rotation 3D ED. Selected reciprocal space layers of single crystal diffuse scattering reconstructed from the experimentally obtained diffraction patterns are shown in Fig. 2 (for reconstruction and data processing routines see Methods section, Supplementary Methods 1 and Supplementary Note 2). The three probes chosen are sensitive to different structural aspects: in a diffraction experiment, neutrons probe exclusively the nuclei, x-rays probe almost exclusively the electron density and electrons probe the electrostatic potential, which depends on the distribution of both the nuclei and the electrons in the sample. Despite these differences, the main diffuse scattering features are similar in all three experiments. In the $hk0$-layer, for example, a continuously curved diffuse line connects the 400, 220 and 040 Bragg reflections, giving a flower-like pattern with higher-order curved features at larger scattering angles. The $hhl$-layer shows streaks parallel to the ⟨110⟩ directions through the 004 Bragg reflection and a series of bracket-like features.

One significant difference that can be observed in reciprocal space is that the features in the electron diffraction data are broadened relative to the corresponding features in the x-ray and neutron data. This effect is seen also for the Bragg reflections, and we attribute the broadening to a combination of setup and sample-related effects. As for the setup-related effects, we attribute the majority of the

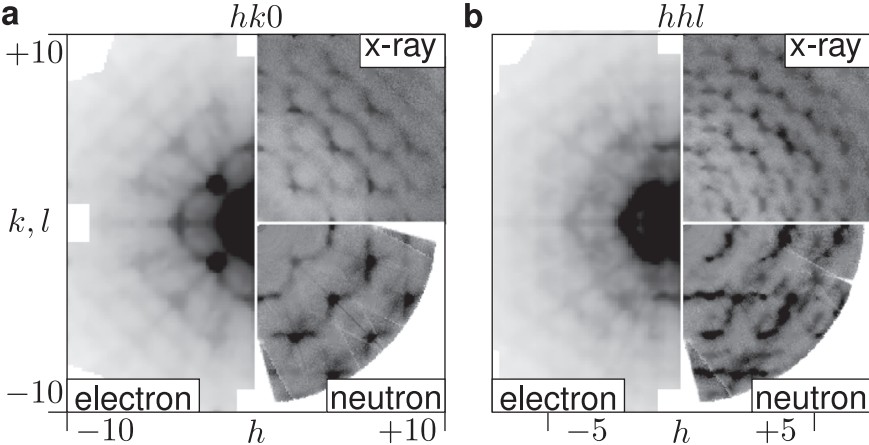

**Fig. 2 | Comparison of reciprocal space sections.** Diffuse scattering in reciprocal space reconstructions after data treatment for x-ray, neutron and electron diffraction experiments. **a** $hk0$-section, **b** $hhl$-section.

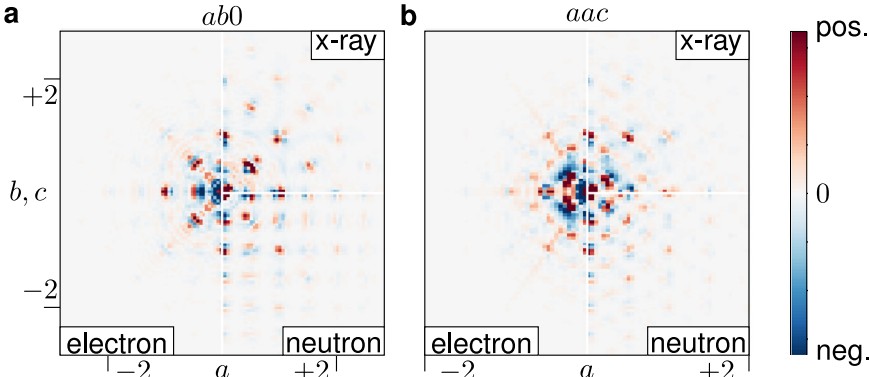

**Fig. 3 | Comparison of 3D-ΔPDF sections.** 3D-ΔPDF sections generated from electron diffraction data compared to the corresponding sections from neutron and x-ray data. **a** $ab0$-plane, **b** $aac$-plane. Positive correlations are in red, and negative correlations are in blue.

broadening in reciprocal space to inelastic scattering: In the centre of reciprocal space a broad background due to inelastically scattered electrons can be observed. This background can be subtracted and is of little concern. However, inelastically scattered electrons that propagate in the direction of the Bragg peak or diffuse scattering will broaden these features, which cannot be easily removed from the resulting diffraction pattern (see Supplementary Discussion 1).

Another reason for the observed broadening of features in the electron diffraction pattern are sample-related effects from the sample preparation and the nanocrystalline nature of the lamella. A focused ion beam (FIB) was needed to prepare a sample thin enough to minimise the contribution of Kikuchi lines even at increased exposure times (see Supplementary Note 1). On the other hand, the FIB may reduce the crystallinity and increase the mosaicity[42]. Despite the consequences in reciprocal space and real space, the structural parameters of interest could be extracted from 3D ED data.

## Real space

To develop a local order model from the diffraction data described in the previous section, we use the 3D-ΔPDF maps as shown in Fig. 3. The first and most striking difference between the electron and other 3D-ΔPDFs here is that the signatures in the electron 3D-ΔPDF maps are much more localised in the centre of the PDF-space as compared to the x-ray and neutron maps. This is a direct consequence of the experimental broadening of the diffuse scattering in reciprocal space: the sharpness of diffuse scattering is inversely related to the correlation

length of the corresponding local correlations in real space (see Supplementary Discussion 1). Another directly observable difference is that the features in the electron 3D-ΔPDF map seem broader than for the x-ray and neutron maps. This fact we attribute to the nature of the probes: For neutron diffraction the atoms are essentially point scatterers. The electron density of single atoms probed by x-ray diffraction has a significant spatial extent on the length scale observed in the shown 3D-ΔPDF maps. It is the electrostatic potential of the atoms that has the greatest spatial extent of the three signals compared here. Hence, the broadest features are expected in the electron diffraction 3D-ΔPDF maps.

For a material with unknown local order, the goal is to identify the short-range deviations from the average structure and to model the very local interactions. The suppression of higher-order correlations does not hinder this goal. We compare the correlations at the shortest interatomic vectors in our 3D-ΔPDF maps with those determined elsewhere using x-ray and neutron 3D-ΔPDFs[23,38] (Fig. 1). Three-dimensional renderings of the 3D-ΔPDF regions that govern the local order model are shown in Fig. 4 (detailed two-dimensional sections are provided in the Supplementary Note 3). In 3D-ΔPDF maps, regions with exclusively positive or negative density are indications for predominant chemical local ordering, while regions with alternating positive and negative density are indications for predominant displacement disorder[20]. Our study was performed at room temperature and we attribute the observed and evaluated displacive signatures primarily to static displacements. This is in line with previous studies

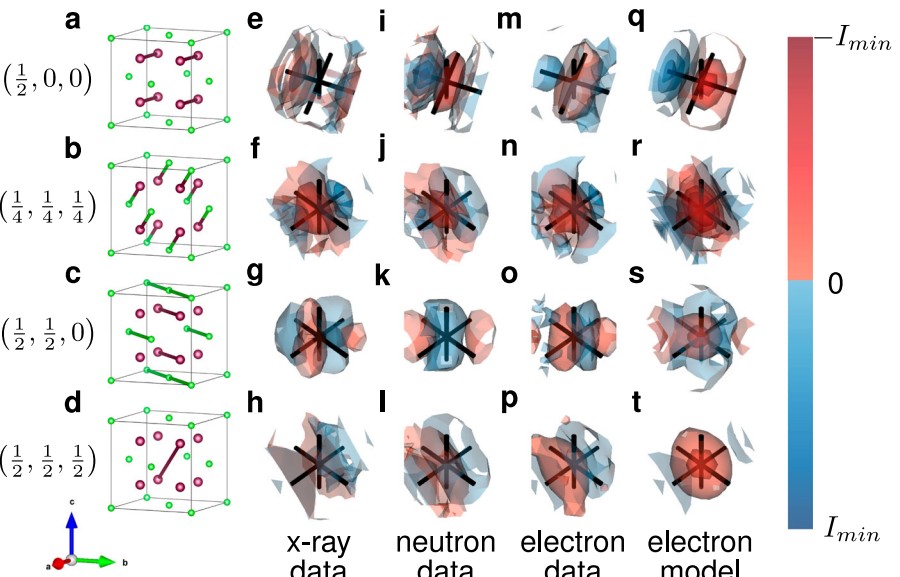

**Fig. 4 | Three-dimensional renderings of the 3D-ΔPDF signatures at the shortest interatomic vectors generated from electron (m–p), neutron (i–j) and x-ray (e–h) diffraction experiments compared to the calculated signatures (q–t).** The leftmost column (a–d) highlights the respective interatomic vectors as links in the average fluorite structure model. Metal sites are green and oxygen (and vacancy) sites are purple. The rightmost column (q–t) shows the signature generated from the calculated diffraction pattern of a simplified model[23] using the kinematic approximation. Black lines in the renderings are along the ⟨100⟩-directions and the crossing indicates the average interatomic vector. Rendering volume in the range −0.15 ≤ Δa, Δb, Δc ≤ 0.15 around the average interatomic vector. Positive correlations are in red, and negative correlations are in blue. Isosurfaces are shown relative to the minimum observed density in the respective 3D-ΔPDF with the lowest isosurface at 5 % (1 % for the $(\frac{1}{2}, \frac{1}{2}, \frac{1}{2})$ interatomic vector).

that showed, that oxygen ion diffusion only becomes relevant at elevated temperatures[43,44].

The ⟨$\frac{1}{2}$00⟩ interatomic vectors are the shortest interatomic vectors that correspond exclusively to difference vectors between oxygen positions. Here and in the following, we use $\Delta^{\pm}_{XY}$ to describe the shift amplitude of the position of a maximum/minimum in the 3D-ΔPDF away from the average interatomic vector between an $X$ and a $Y$ site. The signatures we observe in the electron, neutron and x-ray 3D-ΔPDFs (see Fig. 4 and Supplementary Note 4) all show a minimum shifted by $\Delta^{-}_{OO}$ along [100] towards the centre of PDF-space and a maximum shifted by $\Delta^{+}_{OO}$ along [100] away from the centre of PDF-space. This is consistent with a local relaxation of oxygen ions towards a neighbouring vacancy[20,23].

The ⟨$\frac{1}{4}\frac{1}{4}\frac{1}{4}$⟩ interatomic vectors are the shortest interatomic vectors that correspond to difference vectors between oxygen and metal ion positions. In analogy to the ⟨$\frac{1}{2}$00⟩ interatomic vectors, the signatures we observe from the three radiation types are consistent (see Fig. 4). All signatures show a minimum shifted by $\Delta^{-}_{OM}$ along [111] away from the centre of PDF-space and a maximum shifted by $\Delta^{+}_{OM}$ along [111] towards the centre of PDF-space. This is consistent with a local contraction of the oxygen metal bond, indicating that metal ions adjacent to a vacancy shift away from that vacancy along the body diagonal[23].

The ⟨$\frac{1}{2}\frac{1}{2}$0⟩ interatomic vectors are the shortest interatomic vectors that correspond to difference vectors between metal ion positions, but they also occur as difference vectors in the oxygen substructure. Thus, we observe a superposition of densities resulting from metal-metal interactions and densities resulting from oxygen-vacancy interactions[23]. This superposition is most pronounced in the neutron 3D-ΔPDF, since the neutron scattering length of oxygen is comparable to that of the metals, whereas in x-ray and electron diffraction the metals dominate the scattering process and consequently the densities in the 3D-ΔPDFs (Supplementary Discussion 2 for the quantitative comparison of the scattering factors). This furthermore complicates a potential disentanglement of chemical short-range

order of the metal ions: The difference in scattering length of Y³⁺ and Zr⁴⁺ is marginal for electron and x-ray diffraction experiments – prohibiting a disentanglement of chemical short-range order with the extent of displacement disorder present in YSZ. The neutron scattering lengths of Zr and Y show a small but significant difference but are on the same order as the scattering of O. Hence, also from the neutron 3D-ΔPDF a clear disentanglement of potential chemical metal order is not possible. However, the previously derived local relaxations in the present case allow us to assign the maximum observed at $(\frac{1}{2} + \Delta^{+}_{MM}, \frac{1}{2} + \Delta^{+}_{MM}, 0)$ in the x-ray and electron 3D-ΔPDF to an elongated metal-metal vector resulting from two metal ions that are bridged by one oxygen ion and one vacancy, with both metal ions shifting away from this vacancy[23].

## Quantitative comparison

To evaluate the quantitative reliability of the 3D-ΔPDF, we quantify the shifts of the maxima and minima described qualitatively in the previous section ($\Delta^{+}_{OO}$, $\Delta^{-}_{OO}$, $\Delta^{+}_{OM}$, $\Delta^{-}_{OM}$, and $\Delta^{+}_{MM}$, a detailed description of the fit procedure and its applicability are given in the Supplementary Note 4). We approximate the intensity distributions at the shortest inter-atomic vectors with three-dimensional Gaussian distributions[23]. The refined shift magnitudes are visualised in Fig. 5; a complete list of the parameters including the variances is given in Supplementary Tables 1–3. What is immediately clear is that the refined quantities are similar for all three types of radiation used. This is a key result of our study and demonstrates the viability of electron 3D-ΔPDF approaches.

We do not expect the refined shift magnitudes to be identical for all three radiation types used, as different probes are sensitive to different structural aspects. In particular, for the $(\frac{1}{4}, \frac{1}{4}, \frac{1}{4})$ and $(\frac{1}{2}, \frac{1}{2}, 0)$ vectors in PDF space, the shift magnitude corresponds to an average over the two metal ions. Because the different radiation types have different contrasts in the scattering length of Zr⁴⁺ and Y³⁺, differences in the quantified correlation vectors are therefore to be expected (see Supplementary Discussion 3 for a more detailed discussion of the variations due to the differences in scattering lengths). For the $(\frac{1}{2}, 0, 0)$

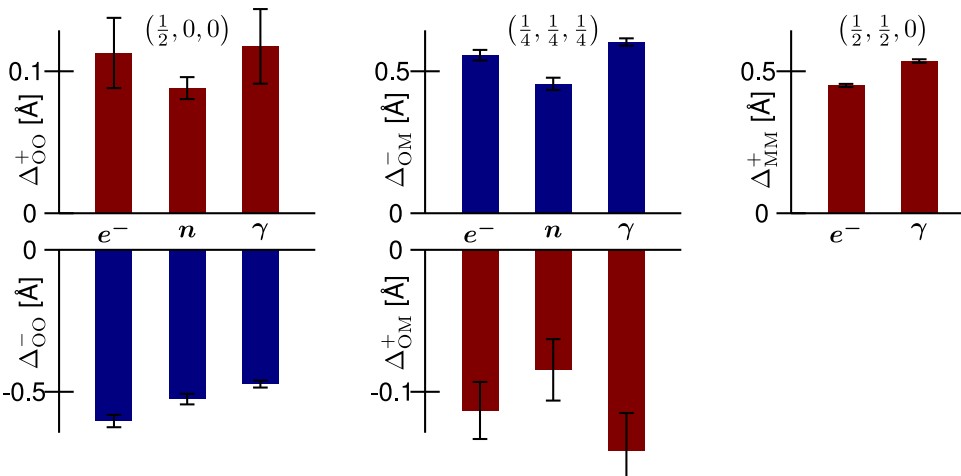

**Fig. 5 | Shifts of the observed maxima (red) and minima (blue) away from the average interatomic distance.** Shifts are estimated by fitting a three-dimensional Gaussian distribution to the data displayed in Fig. 4. Positive shift magnitudes correspond to shifts away from the centre of real space, i.e., elongated interatomic distances. Error bars indicate the $3\sigma$ level of the fit uncertainty. For details see Supplemental Tables 1–3.

vectors on the other hand we expect the neutron value to be the most accurate estimate due to the greater relative scattering power of oxygen compared to the metals (see ref. 23).

### Increased relative sensitivity to lighter elements

The 3D-ΔPDF analysis shows that the most pronounced local oxygen-oxygen interaction at the $\langle\frac{1}{2}00\rangle$ interatomic vectors can be quantified using the x-ray 3D-ΔPDF. The relative scattering strength of oxygen with respect to the metal ions is greater in electron diffraction than in x-ray diffraction (see Supplementary Figure 16 for a graphical visualisation). It is well known from conventional Bragg data analysis that this difference in scattering power enhances the detectability of the lightest elements in electron diffraction experiments compared to x-ray diffraction experiments[45,46]. An excellent example of this relative sensitivity are the 3D-ΔPDF signatures in the vicinity of the $\langle\frac{1}{2}\frac{1}{2}\frac{1}{2}\rangle$ interatomic vectors where the tendency of YSZ to form 6-fold coordinate metals can be observed (compare Fig. 4): The $\langle\frac{1}{2}\frac{1}{2}\frac{1}{2}\rangle$ interatomic vector exclusively occurs on the oxygen sublattice. In the neutron diffraction experiment we observe a weak but clear maximum at this interatomic vector – an indication for positively correlated chemical short-range order[20], i.e. a preference of vacancy pairs along this interatomic vector[23]. In the x-ray data, this signature is unresolved and indistinguishable from residual noise. But in the electron diffraction 3D-ΔPDF, the same signature seen in the neutron data is now clearly identifiable. This observation highlights the increased sensitivity of the electron diffraction experiment to the lighter elements, which can be exploited not only in the analysis of Bragg data analysis but also in the analysis of diffuse scattering.

### Comparison to a computational model

In an earlier study[23], we constructed a local order model based on the x-ray and neutron diffraction data. This model was used as the basis for Monte Carlo simulations that generated an ensemble of $10 \times 10 \times 10$ supercells that captured the experimentally determined local correlations in a simplistic atomistic model. With access to this model, we were able to calculate the expected electron diffuse scattering (by applying the kinematic approximation), and hence the corresponding 3D-ΔPDF[18]. Three-dimensional renderings of the calculated 3D-ΔPDF around the shortest interatomic vectors are shown in Fig. 4 (q–t); further two-dimensional sections that directly compare the calculated and experimentally obtained 3D-ΔPDF are provided in Supplementary Note 5. The computational model reproduces the main features that we analyse using the 3D-ΔPDF from the electron diffraction experiment as well as those from the x-ray and neutron experiments (compare[23]).

### Discussion

We have established that electron diffraction can yield useful 3D-ΔPDF maps from submicron grains, overcoming the single-crystal limitations of x-ray and neutron techniques. The observed broadening of the diffuse scattering in the electron diffraction data does not prevent a quantitative and reliable analysis. This is because the nature of local order is encoded in the position and shape of diffuse scattering, whereas the width of diffuse scattering features relates mainly to correlation length[17,18]. Thus, as we show here for the case of YSZ, it is possible to derive a local order model despite the experimentally observed broadening of the scattered intensities in the electron diffraction experiment. Determining such a local order model is usually the step considered most difficult in the analysis of diffuse scattering. The analysis of the extent of the correlations, which can be derived from the width of the observed features in reciprocal space, is straightforward and can in turn be solved by determining an instrumental resolution function in reciprocal space and hence the maximum observable correlation length[43] (see Supplementary Discussion 1).

A major challenge is the limited reciprocal space coverage available in electron diffraction experiments. The generation of the 3D-ΔPDF from experimentally collected diffuse scattering data requires the application of a three-dimensional Fourier transform[20,22]. Therefore, full reciprocal space coverage is essential in the diffraction experiment. Modern single-crystal x-ray and neutron diffractometers are designed so that this coverage is routinely achieved by measuring only one crystal. Electron diffraction experiments are mostly performed in conventional TEM setups with limited tilt angles[47]. In our case the data were collected from one crystal and a goniometer tilt range of $\pm 50°$. If the symmetry in reciprocal space is high, then the missing wedge in reciprocal space can be filled by symmetry averaging as demonstrated for this study for a structure with cubic symmetry. However, if the symmetry is lower, this option in general does not result in full reciprocal space coverage. In these cases, we suggest combining data from several crystals, which is the typical approach used to increase completeness in 3D ED[48].

Another challenge is the stability of the sample with respect to the electron-beam-induced radiation damage. This may be an issue even in

x-ray diffraction experiments[49,50], but samples in electron diffraction experiments are in general more susceptible to radiation damage due to their small volume. If the material is beam-sensitive, the duration the crystal is stable under exposure additionally limits the amount of data that can be collected from a single crystal. The problem is mitigated by cooling the sample during data collection or working with a cryo-transfer[28,51]. The latter is essential if the vacuum conditions destabilise the sample. In all cases, the combination of data from several crystals makes it possible to achieve full coverage even if individual measurements cover only a fraction of reciprocal space.

As electrons are charged particles, the interaction with matter is much stronger in comparison with the interaction strength of x-rays or neutrons. The probability that a single electron passing through the sample is scattered more than once is significant. The effect of multiple scattering leads to a non-linear dependence of diffracted intensities on, e.g., the crystal thickness and orientation, which is quantitatively described by the dynamical theory of diffraction. It has been shown that taking multiple scattering into account significantly improves the refinements of average structure models against reflection intensities from 3D ED data[45,46,52]. The kinematical approximation assumes that each electron is only scattered once and is routinely applied for more than a decade to solve and refine average structure models[53,54]. To the best of our knowledge, diffuse scattering analysis of 3D ED data – including this work – has always assumed kinematical diffraction theory. Our quantitative analysis, the comparison to the x-ray and neutron experiments, and the agreement with a simple computational model, collectively demonstrate that the kinematic approximation gives reasonable and useful results. We nevertheless expect that based on dynamical diffraction theory the accuracy of diffuse scattering analysis can be improved, which remains to be analysed in future studies.

We further neglect any inelastic scattering, which is partly responsible for the strong background close to the centre of the diffraction patterns. In thicker samples, inelastic scattering gives rise to Kikuchi lines. This must be avoided because the Kikuchi lines would occlude the diffuse scattering features and make the analysis in terms of 3D-ΔPDF unnecessarily challenging or even impossible. The analysed sample of YSZ with a thickness of around 60 nm did not show any signs of Kikuchi lines, and the background was removed during data processing. The effect of inelastic scattering on the resulting models remains to be analysed, either by modelling the inelastic scattering or by working with energy-filtered data. Energy-filtered data were not available for this study.

Another point to consider is the method of detecting the diffuse scattering. As the diffuse scattering intensities are typically $10^3$ to $10^4$ times weaker than the Bragg scattering intensity, the dynamic range of the detector is of paramount importance[17]. In neutron diffraction experiments the neutron flux is the limiting factor in obtaining reliable diffuse scattering data[17]. In x-ray diffraction, large area single photon-counting hybrid pixel detectors are standard in modern synchrotron and laboratory diffractometers[17] and allow fast and almost noise-free data collection. In electron diffraction, it is essential that the detector dynamic range is sufficient to collect reliable diffuse scattering data in the presence of strong Bragg reflections. For this purpose, hybrid pixel detectors like the one used in this study are very useful.

Another point to consider is the validations of the local order model derived from the 3D-ΔPDF analysis – especially in cases where no single crystal x-ray or neutron diffraction experiment is available as was the case demonstrated here. A standard validation approach that – in the opinion of the authors – should always be applied is the realisation of the derived local correlations in a computational model and the subsequent calculation of the diffuse scattering and the 3D-ΔPDF of this model. However, if further information from high-resolution transmission electron microscopy experiments, nuclear magnetic resonance, various spectroscopic methods, x-ray or neutron powder pair distribution function analysis is available this information should

always be included in the modelling process; with the limitation that for the latter techniques the bulk material needs to be phase pure or a multiphase composition needs to be modelled explicitly.

Electron diffraction experiments are routinely performed on submicron crystallites, as these are often synthesis products, whereas large single crystals suitable for x-ray or neutron diffraction experiments often need to be grown by especially adapted synthesis methods[55]. Such a synthesis could potentially alter the local order properties. The use of 3D-ΔPDFs from electron diffraction experiments will allow the characterisation and quantification of local order phenomena in novel functional disordered materials, without the need to adapt the synthesis for the diffraction experiment. Furthermore, the use of electrons as a probe in the diffraction experiments improves the simultaneous detectability of light and heavy elements as compared to x-ray diffraction[45,46]. This is crucial for functional oxide materials, such as our reference material YSZ, where both metal and oxygen ions are disordered. In these cases, the use of electrons as a probe allows the detection of weaker correlations of lighter elements.

Powder PDF is and will remain an important characterisation method, but its reduced information content can lead to difficulties and ambiguities of interpretation that are avoided with 3D methods. A key advantage of powder PDF, however, is that it easily allows in-situ, in-operando and variable pressure experiments, which currently can only be implemented to a very limited extent in a 3D ED setup. We envisage that many studies will now benefit from combining both approaches. For example, a three-dimensional local order model from 3D ED in conditions accessible to electron diffraction can provide a starting point for the adaption of the model that then describes the PDF data measured at the conditions of interest. We consider this combination to be the optimal use of the proposed method and we see great potential in its application to solving complex disorder problems.

## Methods

### Sample material and preparation
The zirconia samples have a composition of $Zr_{0.82}Y_{0.18}O_{1.91}$, grown by the skull melting method, delivered by Djevahirdjan S. A., Monthey, Switzerland. The composition was confirmed by energy-dispersive X-ray spectroscopy (EDX)[23]. For neutron measurements, the large, clear single crystals were cut with a diamond saw to cubes with an edge length of approximately 5 mm. For X-ray diffraction measurements the larger crystals were mechanically ground to a diameter of about 150 μm and polished. For electron diffraction measurements a small grain of the crystal was attached to a lift-out grid (SPI Supplies Omniprobe) and an area of about 8.6 μm by 3.9 μm was thinned to a final thickness of about 40 nm to 60 nm using a focused beam of Ga ions (FEI Quanta 3D FEG). More information is provided in Supplementary Note 1.

### X-ray diffraction measurements
X-ray experiments were performed on a Rigaku Synergy S diffractometer equipped with an Eiger 1 M detector using Mo radiation, $(\sin(\theta_{max})/\lambda = 1.28 \text{ Å}^{-1})$. To avoid possible fluorescence a threshold of 17.4 keV was used on the detector. Simple $\omega$-scans with 0.5° step widths and 120 s exposure time were taken. The crystal was kept at ambient conditions. 3D diffuse scattering data was reconstructed on a $501 \times 501 \times 501$ voxel reciprocal space grid ($-10 \le h, k, l \le 10$) using the orientation matrix provided by CrysAlis Pro[56] and custom Python scripts using Meerkat[57].

### Neutron diffraction measurements
Neutron diffraction experiments were carried out at D19 instrument ($\lambda = 0.95$ Å, 0.1° steps, 80 s exposure per frame, $\sin(\theta_{max})/\lambda = 0.94 \text{ Å}^{-1}$), Institut Laue-Langevin (ILL), Grenoble utilising a 180° $\phi$-scan. 3D diffuse scattering data reconstruction utilised the orientation matrix as

provided by Int3d[58] and a custom Python script. The data was reconstructed on a $501 \times 501 \times 501$ voxel reciprocal space grid ($-10 \leq h, k, l \leq 10$).

## Electron diffraction measurements

Electron diffraction experiments were carried out with an FEI Tecnai G2 microscope using 200 keV electrons ($\lambda = 0.02508$ Å, $\sin(\theta_{\max})/\lambda = 1.24$ Å$^{-1}$) equipped with a hybrid-pixel detector ($512 \times 512$ pixels, Cheetah from Amsterdam Scientific Instruments ASI). A condenser aperture with a diameter of 10 μm and spot size 9 was used to optimise the electron beam for data collection of a small area (microbeam diffraction mode with a beam diameter of about 1.5 μm). A selected area aperture was not used. Diffraction patterns were taken with 1.512 s exposure time and the crystal was rotated by 0.4° during each exposure. The goniometer was tilted from −50° to +50° and 200 patterns were recorded. PETS2[59] was used to refine the orientation and centre of the frames, and to export the 3D diffuse scattering reciprocal space map. The data was reconstructed on a $201 \times 201 \times 201$ voxel reciprocal space grid ($-10 \leq h, k, l \leq 10$). Due to the lower detector resolution and the observed broadening of the diffracted features this smaller grid was chosen for the electron diffraction experiment as compared to the neutron and x-ray experiments. The effect of the binning size of the data reconstruction is discussed in more detail in Supplementary Note 4.

## Data treatment procedures

The reflection conditions for the *F*-centring were fulfilled in all cases and after careful inspection, the data were symmetry-averaged in the $m\bar{3}m$ Laue symmetry. The general data processing procedure to obtain 3D-ΔPDF from experimental data is described in ref. [60]. The experimentally obtained data were treated with the KAREN outlier rejection algorithm[61] and additionally a custom punch-and-fill approach that interpolates the intensity in punched voxels was used to eliminate residual Bragg intensities. To avoid Fourier ripples the data were multiplied with a Gaussian falloff that smooths the edges of the measured reciprocal space sections (see[61]). The fast Fourier transform algorithm as implemented in Meerkat[57] was used to obtain 3D-ΔPDF maps.

# Data availability

The experimental raw data, reciprocal space reconstructions and 3D ΔPDFs are deposited and available at https://doi.org/10.5281/zenodo.8207506 (and https://doi.org/10.5291/ILL-DATA.5-13-277 for neutron diffraction raw data).

# Code availability

All custom code used in this study was developed using widely available algorithms. Copies of the code used can be obtained upon reasonable request.

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

## Acknowledgements

Reinhard B. Neder (University of Erlangen-Nuremberg), Arianna Minelli (Oxford University and Oak Ridge National Lab) and Marie-Hélène Lemée (ILL, Grenoble) are thankfully acknowledged for providing the sample material, assisting in sample preparation and conducting the neutron diffraction experiment. We thankfully acknowledge Mariana Klementova for the help in the preparation of the sample for electron diffraction. The work presented here was supported by the Czech Science Foundation (grant number 21-05926X to L.P.) and by the European Research Council (advanced grant 788144 to A.L.G.). CzechNanoLab project LM2023051 funded by MEYS CR is gratefully acknowledged for the financial support of the measurements/sample fabrication at LNSM Research Infrastructure.

## Author contributions

E.M.S., P.B.K., Y.K., A.L.G. and L.P. designed the research. P.B.K., Y.K. and P.S. prepared the material and conducted the electron diffraction experiments. P.B.K. and E.M.S. performed the data treatment. E.M.S. performed the 3D-ΔPDF analysis and the model calculations. E.M.S. wrote the manuscript, with input from all authors.

## Funding

## Competing interests

The authors declare no competing interests.
