## [Peer Review File · Nature Communications]

REVIEWER COMMENTS

Reviewer #1 (Remarks to the Author):

This manuscript by Schmidt et al demonstrates the use of diffuse electron scattering to probe local structural ordering in yttria-stabilized zirconia. The results are compared to some authors' recent work with diffuse x-ray scattering and diffuse neutron scattering (Schmidt et al, Acta. Cryst. B 79, 2023). Yttria-stabilized zirconia has been extensively studied already (with proper citation in this manuscript), with the previous work in Acta Cryst. demonstrating the value of the newer 3D- Δ PDF approach. The main innovation in this manuscript is the use of electrons to achieve a full reciprocal space volume (as opposed to oriented sections) suitable for 3D- Δ PDF analysis.

This is a nice paper, and the proposed application of diffuse electron scattering to smaller crystals may help open up the study of short-range order to a broader community. I have a few comments and questions:

1) The introductory discussion on interpretation of 3D- Δ PDF (lines 59-70), while technically correct, is difficult to follow. It might help to more clearly distinguish between a correlation and its signal in the 3D- Δ PDF (these two can be identified with one another, but it reads as a circular definition).

2) The presented analysis is very much focused on fitting relaxations around vacancies, but the presence or absence of chemical short-range order needs to be mentioned more clearly. Clear statements about ordering or the absence of ordering between Zr/Y atoms and between metal atoms and vacancies would be helpful for those not already familiar with the system; more importantly, the presence of chemical SRO will add additional a peak to the very regions of the 3D- Δ PDF, potentially complicating the fitting analysis done here. The authors do state that vacancy-vacancy pairs at $\langle \frac{1}{2}\frac{1}{2}\frac{1}{2} \rangle$ vectors are more clearly observed with electrons than neutrons, which is a good finding, but it could use a more rigorous statement. Ordering of Zr/Y atoms and expected oxygen vacancies near each type of metal atom should be clearly stated as well. In the absence of such a statement, Figure 1 seems to imply that Zr and Y tend towards short-range order, while Figure 4 subtly suggests the distribution of Zr/Y atoms is completely random through its fractional coloring scheme (probably not intentional?). This issue need not take over the paper, but a few clear and well-justified statements on chemical ordering (not just in references) are needed to justify the work on displacements. If chemical ordering can be found directly in the measured electron scattering, all the better!

3) Experimental broadening due to "particular setup" is vague – x-ray and neutron data had a similar resolution to each other (at least in binning); why is the resolution less fine for the electron measurements? I might guess some combination of incident electron energy, detector pixel size, and detector distance for the instrument limited resolution to less than a typical x-ray experiment, but are these truly particular to this setup, or more inherent to electron instruments broadly? Could a currently available electron setup achieve similar resolution to the x-ray and neutron data shown? A long general discussion is not necessary, but a bit more discussion on why this particular setup uses a larger bin size in reciprocal space would be valuable for those not familiar with electron scattering.

4) Figure 5 presents the results of quantitative fitting, but these quantities are not quite the same as expected displacements in the system for each kind of intersite pair -- the scattering lengths are different for each probe, but the underlying distributions of displacements are the same. Is it possible to infer the expected displacements from the observed shifts? If not, could one make qualitative guesses as to the relative values of the shifts as measured by different probes and show that they match what is fitted? Also, how much does it matter that the positive/negative pairs are not equally displaced from the average position? Part of this might be related to item (2) above.

5) A more open-ended question: how much of the observed scattering can be attributed to displacement correlations similar in form to thermal diffuse scattering? Some of the figures (the neutron $\langle \frac{1}{2}\frac{1}{2}0 \rangle$ in Fig. 4 is an example) resemble to displacement correlation signatures. This might just need a brief explanation, but this may also affect interpretation of the extracted shifts.

To sum up: I think that the experimental procedure presented here is valuable and interesting, and I think that some clarification on exactly what is being extracted from the data in the analysis could be very helpful to a community that is slowly getting more interested in short-range order. I'd like to make sure that the key points are all being as clearly communicated as possible before

the manuscript is published.

Reviewer #2 (Remarks to the Author):

In my (not completely unbiased) perception, there have been two topics in the post-pandemic crystallographic conferences that have experienced a particularly strong increase in interest: 3D electron diffraction (3D-ED) and the 3D-DeltaPDF. This is also reflected in a number of papers published in leading scientific journals in recent years. A few of them are cited in this paper. In the present study, the authors show how 3D-ED and 3D-DeltaPDF methods can be successfully combined and open up new possibilities to reliably characterize local structure of disordered structures that have so far eluded detailed investigation: crystallites that have too complex (disorder) structures to be analyzed by powder methods and are too small to be studied by single crystal X-ray methods (or had to be measured at one of the few synchrotron beamlines highly specialized for sub-micro crystals). The authors commendably took the trouble to measure the same structure with X-ray, neutron, and electron diffraction methods and to prepare the data in such a way that 3D-DeltaPDF maps could be successfully calculated in each case. The comparison of the different measurements was complicated by the fact that the respective methods produce different artifacts and have different structural sensitivities. Nevertheless, the authors succeeded in convincingly demonstrating that the analyses for the different methods provide at least semiquantitatively comparable results and that 3D-ED experiments may become a strong complement to X-ray and neutron-based 3D-DeltaPDF investigations.

Also, the advantages, disadvantages, and limitations of the different methods are compared and discussed in detail. Below, I will discuss some further limitations of the study, which, however, are not unusual for such an early proof-of-concept study and do not diminish the importance of this paper in any way. On the contrary, I expect that this paper will stimulate further important research. I see no reason why the current limitations cannot be overcome in the near future by more experience and improvement of the available tools, especially since the development of such tools is already in full swing. In the case of 3D-ED, significant progress is currently being made in instrumentation and handling of dynamic scattering effects, while in the 3D-DeltaPDF community a great deal of experience is currently being gained in understanding complex disordered structures and the necessary software is being developed for easier access to the disordered information. The current development of 3D-DeltaPDF tools is focused on X-ray applications, but may straightforwardly be applied to 3D-ED data or easily be adapted. There is no doubt that the 3D-ED / 3D-DeltaPDF combination will benefit greatly from all these developments and will quickly establish itself as an important material characterization tool. I think this paper should therefore definitely be published in nature communication.

My only substantive criticism, which I think requires an adjustment of the manuscript, is the comparison of the measurements of the positions of some 3D-DeltaPDF signals and the conclusion (partly explicit, partly suggested) that the three methods provide quantitatively comparable disorder models. Here I do not fully agree. First, some results show very strong discrepancies, especially in the data shown in the Supplementary material. There may be good reasons for this due to the different measurement methods, but they had better be rationalized if used as an argument for quantitative comparability of the results from the three measurements. On the other hand, for small displacements, the position of a 3D-DeltaPDF signal is not a good measure of the local displacement correlations. On the contrary, it is almost exclusively the strength of the signal and not the location that is decisive when, for example, two Gaussian functions are subtracted from each other, which is an important feature of 3D-DeltaPDF models. Unfortunately, I am not aware of any literature reference that addresses this, but I have included a plot for demonstration purposes that should be self-explanatory. In other words, a good match in the positions of 3D DeltaPDF signals is not a proof that full quantitative modelling would also give comparable results for displacement correlations. But this would be what we are interested in. Unfortunately, tools for quantitative modelling of 3D-ED / 3D-DeltaPDF models are not available at the moment, so quantitative modelling cannot be required from the authors. However, the text should be worded accordingly a bit more carefully to not suggest wrong conclusions.

The further suggestions for improvement are in the broader sense of linguistic or stylistic nature

and I leave it to the authors to accept these or not.

- The term 'correlated disorder' is used several times. I find this somewhat unfortunate, since 'disorder' is more a global descriptor ('disorder model', 'disordered structure', etc., also the term 'disordered atom' is more associated with average structure properties) and less about very localised properties, which are seen in the 3D-DeltaPDF. I would prefer a continuous use the term 'local order' as a more accurate description.
- In the second line of the abstract it is written about 'intentional introduction of disorder'. I think this is a bit unfortunate, since unintentional disorder can be just as interesting.
- line 34ff: It would be good if some non-perovskite examples were also listed.
- line 158: new symbols Δ^{\sim} etc. are introduced without defining them. In the course of the text the meaning became clear to me, but without explanation they could be misunderstood.
- Fig. 4: I cannot distinguish the colors for Zr and Y. Maybe a better color coding would be helpful.
- line 179: In the context of 3D-DeltaPDF features I would speak of 'densities' instead of 'intensities', because they are derived from scattering densities.
- lines 218 - 235: It is repeated several times that ED is more sensitive to light elements than X-ray diffraction, which affects the readability of this paragraph. Consider rewriting this paragraph.
- Chapter 'Challenges ...' The challenge of multiple scattering in ED should be included in this paragraph. An additional paragraph discussing the possible influence of multiple scattering on the 3D-DeltaPDF would certainly be interesting, in particular, since some of the authors already have a lot of experience regarding average structure refinements based on 3D-ED diffraction. Would this experience be transferable to diffuse scattering?
- Supplement chapter 4: I would write of 'dashed lines' instead of 'dotted lines' and again replace 'intensities' with 'densities'. Also one should write that the crossing points of the lines describe the average interatomic vector. In S4 there are four crossing points which should be listed in the caption.

Reviewer #3 (Remarks to the Author):

Dear Authors

The present study describes a successful use of electron diffraction data for conducting 3D local structure analysis on an inorganic material of technological interest, namely yttria stabilized zirconia. The general goal is to demonstrate that the analysis of local structure such as the spatial arrangement of defects and displacive disorder can be determined reliably and in 3D by electron diffraction, thus overcoming crystal size limitations imposed by single crystal X-ray and neutron diffraction. In this respect, this study presents a convincing use of electron diffraction data for conducting 3D-deltaPDF analysis, while conveniently comparing it with analogous results from X-ray and neutron 3D-deltaPDFs that have been previously published.

Overall, the study conveys an important message and highlights prominent challenges that need to be solved to make this technique more robust and widely used. As for the "quantitative" aspects of it, particularly standing out in the title and the manuscript itself, I believe that this study should put forward some more caveat and cautionary considerations. In the context of modern crystallography and materials science, this work constitutes an important step beyond the state of the art, which undoubtedly qualifies it for publication. I do have a few points, though, which in my view require attention and I will elaborate in the following. I therefore consider this work valuable as a Nature Communications article after some minor revisions are addressed.

1. Line 41. The Authors mention that the analysis of Bragg diffraction is well established. I would also add that is even "automated", since less and less is left to humans in current crystallographic routines, and crystal structures can be solved by essentially autonomous software (such as the Autochem plugin by Rigaku) when data quality is not particularly problematic.

2. Line 68. The Authors state that the average interatomic vector of neighbouring atoms that vibrate in-phase "is smeared out by the average structure atomic displacement parameters (ADPs)". I understand the intended meaning, but ADPs are used to model electron density distribution, therefore they cannot have any smearing effect themselves. I assume the Authors intended something like "is smeared out in the average structure, as modelled by atomic displacement parameters (ADPs)". For this reason, I think this sentence should be rephrased to make it more accurate.

3. Line 73. I would be more explicit on the crystal size, since "macroscopic" has different meanings depending on the context. Perhaps the Authors can consider using the word "micrometric", and preferably pairing it with "currently" and "in general" to be on the safe side, since sub-microcrystal XRD can be performed in numerous synchrotrons worldwide.

4. The Authors correctly point out that the importance of analyzing nanocrystals is primarily due to the fact that their correlated disorder might be different from their microcrystalline analogues. However, I find this aspect addressed very little when describing the interpretation of the data. For example, at line 130 the Authors describe the broadening of electron diffuse scattering data compared to the X-ray and neutron diffraction one, attributing it to the experimental setup simply based on the fact that both Bragg and diffuse scattering features are affected by broadening at the same time. I believe there are numerous microstructural characteristics that can differ between large microcrystals and nanocrystalline samples, affecting both diffuse scattering and the Bragg peaks' profiles. This is not only a matter of short-ranged correlated disorder, but can also relate to coincidental change in mosaicity and crystallite size. Such effects can also be found in the PDF space, which would potentially invalidate the statement at line 146: "This is a direct consequence of the experimental broadening of the diffuse scattering". I agree that broadening of diffuse scattering relates to correlation lengths in real space, but I am concerned with the use of "experimental" as in 'due to the experimental setup' (meaning in particular all the set of distortions that come with an electron diffraction experiment). I think this is an important aspect to clarify, and personally I would keep the option of having real structure effects impacting patterns and PDFs, which are therefore not due to experimental setup, while having analogous effects. Addressing these aspects could also give the Authors a good chance to comment on the challenge of discriminating the two sources of broadening (structural changes vs. 'experimental' broadening), and the need of quantifying the instrumental sources of distortions in total scattering data from electron diffraction, as well as their effects in the PDF space. This aspect should also be kept in mind when, further in the manuscript, the Authors consider the quantitative differences of correlation lengths. In lines 202 and 205–207, these differences are attributed to the fact that probing matter with electrons and with X-rays or neutrons provides different lengths because of different sensitivity to "structural aspects". However, I would also specify that differences due to the nanocrystalline nature of the sample and its different, unknown microstructure can also be a cause of such varying correlation lengths.

5. The proposed PDF analysis is useful and in line with what previously published based on X-ray and neutron data, but there is an aspect that I did not find addressed explicitly: the impact of vacancy defects on the PDF space. The interpretation appear focused on the consequences of vacancies on structural distortions, leaving the reader with the impression that the only information available from the 3D-deltaPDFs is displacive disorder, and no localisation and quantification of correlation signals due to lack of density (presence of defects) can be distinguished. This might indeed be true, and I believe that, in case, the Authors should state this clearly to better guide the readers. It would be useful to mention qualitatively and briefly what kind of signals can be expected when vacancy defects are present and why they are not interpretable from the available experimental delta-PDFs of nanocrystalline yttria stabilized zirconia.

6. The Authors address the quantitative reliability of the 3D-deltaPDF analysis by comparing

correlation lengths in terms of positions of maxima and minima between X-ray, neutron and electron PDFs. I find this a relatively weak part that could be improved, and I will elaborate in the following.

In my view, there are two aspects that can be considered concerning 'quantitative reliability' in the PDF space: the positions of the correlation vectors, and the numerical values (maxima/minima or integrals) of the correlation signals. It appeared to me that the Authors focus selectively on the first aspect. Nonetheless, the second, too, contains important defect structure information, which can be made less reliable due to artefacts from data processing, instrumental broadening, or multiple elastic and inelastic scattering. It is not possible to use comparisons with XPDFs or with neutron PDFs for assessing quantitative reliability, because both shift magnitudes and integrals at specific correlation vectors can be influenced by the fact that nanocrystals can have different local structures, and because, as the Authors pointed out, "different radiations are sensitive to different structural aspects". For this reason, I believe that using such comparisons to evaluate the quantitative reliability does not help much, and indeed this section concludes with the Authors acknowledging that differences are "to be expected".

I consider these comparisons nonetheless useful, but I wonder whether the Authors could be more transparent with respect to this ambiguity in quantitative reliability, and formulate how it could be determined more appropriately. This remains an open question that does not stand out enough, in my opinion. It could also become an important point to be presented in the "challenges that need addressing" section: how can we validate our 3D-deltaPDF from electron diffraction data, since we cannot compare it with neutron and X-ray PDF analysis due to the different nature of the beam-sample interaction and the possibly different local structure?

7. The supplementary material is missing the pristine reciprocal space reconstructions of the main planes of the sample ($hk0$, $h0l$, $0kl$). Since the production of diffuse scattering reconstructions required not only a reasonable symmetry averaging, but also other steps such as "Karen" filtering, punch-and-fill, and gaussian-dampening, it would be most informative for the readers—especially those who are willing to conduct similar analyses in the future—to have an idea of the pristine reconstruction and possibly some intermediate steps of the data processing. Furthermore, punch-and-fill procedures have also an important effect on the features observed in the 3D-deltaPDF, especially the choice of punch radius. Indeed, different punch radii may change the integrals and shift of correlation peaks, as the Patterson function might partly contribute, which is a likely possibility when the diffuse scattering is primarily originating from correlated distortions in the structure. Ideally, the chosen punch radius should be justified, and the effect of different punch radii could be briefly shown to give an idea of how much this process affects the features that are discussed in the manuscript. This would also provide useful insights on the quantitative side of this analysis, since it is a central aspect the Authors aimed at.

8. When addressing the challenges of the method, the Authors describe three main points: limited reciprocal space coverage, sample stability to the electron beam, and dynamic range of the detector. However, I think an important aspect is missing: the presence of non-kinematical elastic or inelastic scattering. Kikuchi diffraction and other inelastic scattering events can produce broad band-like intensities that might be problematic to distinguish and separate from the data, adding undesired features to the 3D-deltaPDFs. This is particularly relevant for inorganic samples such as the one addressed in this study, where the Authors took care of preparing it as a thin lamella. The strong influence of sample thickness, ubiquitous presence of undesired multiple and inelastic scattering, and the possibility of using energy-filtered diffraction to at least eliminate most of these intensities, would all be important aspects to be discussed in this last section.

Furthermore, the challenge of validating the electron 3D-deltaPDF analysis with complementary techniques for the reasons described above can also be proposed as an open challenge, since there is no standard practice, as far as I am aware, for this purpose.

9. I did not find, in the cited literature, the use of electron diffraction data for 1D PDF analysis, as reported by Gorelik et al. in 2015 (*Microscopy and microanalysis*, 2015, 21(2), 459–471) and later improved towards a quantitatively reliable technique (Gorelik et al. *Acta Cryst.* 2019, B75, 532–549). This is not only highly relevant to this study, but it might also allow a better quantitative reliability assessment of the electron 3D-deltaPDF analysis by comparing the electron 3D-deltaPDF

with its 1D analogue, which, in turn, can be more reliably compared to 1D XPDF.

10. Just a couple of writing slips I noticed. Line 44: Since the subject is "powder pair distribution function analysis", I believe the subsequent verb should be "has been". In the caption of Figure 4, line 175 reads "diffraction patterns a simplified model". I imagine "from" should be inserted before "a simplified model".

Finally, I would like to congratulate the Authors on this beautiful work and wish them all the best for the revision process.

RESPONSE TO REVIEWERS' COMMENTS

Reviewer #1:

This manuscript by Schmidt et al demonstrates the use of diffuse electron scattering to probe local structural ordering in yttria-stabilized zirconia. The results are compared to some authors' recent work with diffuse x-ray scattering and diffuse neutron scattering (Schmidt et al, Acta. Cryst. B 79, 2023). Yttria-stabilized zirconia has been extensively studied already (with proper citation in this manuscript), with the previous work in Acta Cryst. demonstrating the value of the newer 3D- Δ PDF approach. The main innovation in this manuscript is the use of electrons to achieve a full reciprocal space volume (as opposed to oriented sections) suitable for 3D- Δ PDF analysis.

This is a nice paper, and the proposed application of diffuse electron scattering to smaller crystals may help open up the study of short-range order to a broader community. I have a few comments and questions:

1. The introductory discussion on interpretation of 3D- Δ PDF (lines 59-70), while technically correct, is difficult to follow. It might help to more clearly distinguish between a correlation and its signal in the 3D- Δ PDF (these two can be identified with one another, but it reads as a circular definition).

The wording was adapted to avoid the term correlation in the description of the PDF signal. The term correlation is now only used with respect to structural correlations causing the observed diffuse scattering. We now use consistently the terms density and intensity to describe the features in the 3D- Δ PDFs.

2. The presented analysis is very much focused on fitting relaxations around vacancies, but the presence or absence of chemical short-range order needs to be mentioned more clearly. Clear statements about ordering or the absence of ordering between Zr/Y atoms and between metal atoms and vacancies would be helpful for those not already familiar with the system; more importantly, the presence of chemical SRO will add additional a peak to the very regions of the 3D- Δ PDF, potentially complicating the fitting analysis done here. The authors do state that vacancy-vacancy pairs at $\langle 1/2, 1/2, 1/2 \rangle$ vectors are more clearly observed with electrons than neutrons, which is a good finding, but it could use a more rigorous statement. Ordering of Zr/Y atoms and expected oxygen vacancies near each type of metal atom should be clearly stated as well. In the absence of such a statement, Figure 1 seems to imply that Zr and Y tend towards short-range order, while Figure 4 subtly suggests the distribution of Zr/Y atoms is completely random through its fractional coloring scheme (probably not intentional?). This issue need not take over the paper, but a few clear and well-justified statements on chemical ordering (not just in references) are needed to justify the work on displacements. If chemical ordering can be found directly in the measured electron scattering, all the better!

We agree that in the former version of the manuscript the discussion of potential chemical short-range order was too short. We added clarifications about the dominance of relaxations for YSZ as a model system and commented on potential local order for the evaluated interatomic vectors in the revised version. Specifically, sentences (marked in yellow in the resubmitted manuscript) were added to the Introduction (page 6), Results (page 10, page 12), and the Discussion (page 15).

3. Experimental broadening due to “particular setup” is vague – x-ray and neutron data had a similar resolution to each other (at least in binning); why is the resolution less fine for the electron measurements? I might guess some combination of incident electron energy, detector pixel size, and detector distance for the instrument limited resolution to less than a typical x-ray experiment, but are these truly particular to this setup, or more inherent to electron instruments broadly? Could a currently available electron setup achieve similar resolution to the x-ray and neutron data shown? A long general discussion is not necessary, but a bit more discussion on why this particular setup uses a larger bin size in reciprocal space would be valuable for those not familiar with electron scattering.

We added further information on experimental broadening related to inelastic scattering and sample preparation in the main text (Results, page 8 and 9).

To address the issue of the binning size we added in the SI section 6 the respective fits for the X-ray and neutron data also binned to a grid of 201x201x201 voxels to show that the binning introduces no deviations that are larger than the estimated standard deviations from the shifts for these most local correlations examined here.

4. Figure 5 presents the results of quantitative fitting, but these quantities are not quite the same as expected displacements in the system for each kind of intersite pair – the scattering lengths are different for each probe, but the underlying distributions of displacements are the same. Is it possible to infer the expected displacements from the observed shifts? If not, could one make qualitative guesses as to the relative values of the shifts as measured by different probes and show that they match what is fitted? Also, how much does it matter that the positive/negative pairs are not equally displaced from the average position? Part of this might be related to item (2) above.

As it turns out, the shift magnitudes based on the same model in fact depend on the probe used, i.e., the shift amplitudes differ in 3D- Δ PDF maps for X-rays, electrons, and neutrons. We included a discussion of this aspect in the supporting information in Section 9 on page 19. Figure S15 was added to the same section and demonstrates the discussed effect by extracting the shift magnitudes from simulated model data that was calculated from the identical model crystal.

5. A more open-ended question: how much of the observed scattering can be attributed to displacement correlations similar in form to thermal diffuse scattering? Some of the figures (the neutron $\langle \frac{1}{2} \frac{1}{2} 0 \rangle$ in Fig. 4 is an example) resemble to displacement correlation signatures. This might just need a brief explanation, but this may also affect interpretation of the extracted shifts.

We attribute the majority of our observed correlations to displacement correlations of static rather than dynamic origin. Temperature dependent measurements or energy discriminating neutron diffraction measurements could help to resolve the question of how much thermal diffuse scattering contributes to the experimentally observed diffuse scattering. Previous studies showed that diffuse scattering due to oxygen mobility only starts at elevated temperatures and therefore, we do not take this into account here. We clarified this by adding new references 43 and 44 on page 10 (line 192) of the updated manuscript.

To sum up: I think that the experimental procedure presented here is valuable and interesting, and I think that some clarification on exactly what is being extracted from the data in the analysis could be very helpful to a community that is slowly getting more interested in short-range order. I'd like to make sure that the key points are all being as clearly communicated as possible before the manuscript is published.

We thank Reviewer #1 for their detailed input and hope that in the revised version we communicate the key points as clearly as possible.

Reviewer #2 (Remarks to the Author):

In my (not completely unbiased) perception, there have been two topics in the post-pandemic crystallographic conferences that have experienced a particularly strong increase in interest: 3D electron diffraction (3D-ED) and the 3D-DeltaPDF. This is also reflected in a number of papers published in leading scientific journals in recent years. A few of them are cited in this paper. In the present study, the authors show how 3D-ED and 3D-DeltaPDF methods can be successfully combined and open up new possibilities to reliably characterize local structure of disordered structures that have so far eluded detailed investigation: crystallites that have too complex (disorder) structures to be analyzed by powder methods and are too small to be studied by single crystal X-ray methods (or had to be measured at one of the few synchrotron beamlines highly specialized for sub-micro crystals). The authors commendably took the trouble to measure the same structure with X-ray, neutron, and electron diffraction methods and to prepare the data in such a way that 3D-DeltaPDF maps could be successfully calculated in each case. The comparison of the different measurements was complicated by the fact that the respective methods produce different artifacts and have different structural sensitivities. Nevertheless, the authors succeeded in convincingly demonstrating that the analyses for the different methods provide at least semiquantitatively comparable results and that 3D-ED experiments may become a strong complement to X-ray and neutron-based 3D-DeltaPDF investigations.

Also, the advantages, disadvantages, and limitations of the different methods are compared and discussed in detail. Below, I will discuss some further limitations of the study, which, however, are not unusual for such an early proof-of-concept study and do not diminish the importance of this paper in any way. On the contrary, I expect that this paper will stimulate further important research. I see no reason why the current limitations cannot be overcome in the near future by more experience and improvement of the available tools, especially since the development of such tools is already in full swing. In the case of 3D-ED, significant progress is currently being made in instrumentation and handling of dynamic scattering effects, while in the 3D-DeltaPDF community a great deal of experience is currently being gained in understanding complex disordered structures and the necessary software is being developed for easier access to the disordered information. The current development of 3D-DeltaPDF tools is focused on X-ray applications, but may straightforwardly be applied to 3D-ED data or easily be adapted. There is no doubt that the 3D-ED / 3D-DeltaPDF combination will benefit greatly from all these developments and will quickly establish itself as an important material characterization tool. I think this paper should therefore definitely be published in nature communication.

My only substantive criticism, which I think requires an adjustment of the manuscript, is the comparison of the measurements of the positions of some 3D-DeltaPDF signals and the conclusion (partly explicit, partly suggested) that the three methods provide quantitatively comparable disorder models. Here I do not fully agree. First, some results show very strong discrepancies, especially in the data shown in the Supplementary material. There may be good reasons for this due to the different measurement methods, but they had better be rationalized if used as an argument for quantitative comparability of the results from the three

measurements. On the other hand, for small displacements, the position of a 3D-DeltaPDF signal is not a good measure of the local displacement correlations. On the contrary, it is almost exclusively the strength of the signal and not the location that is decisive when, for example, two Gaussian functions are subtracted from each other, which is an important feature of 3D-DeltaPDF models. Unfortunately, I am not aware of any literature reference that addresses this, but I have included a plot for demonstration purposes that should be self-explanatory. In other words, a good match in the positions of 3D DeltaPDF signals is not a proof that full quantitative modelling would also give comparable results for displacement correlations. But this would be what we are interested in. Unfortunately, tools for quantitative modelling of 3D-ED / 3D-DeltaPDF models are not available at the moment, so quantitative modelling cannot be required from the authors. However, the text should be worded accordingly a bit more carefully to not suggest wrong conclusions.

We acknowledge the concern about the derived shift magnitudes and thank the reviewer for raising this important point. We argue that in the special case of YSZ this procedure is justified. In YSZ we assume static displacements of the oxygen ions towards neighbouring vacancies. With this in mind the “average” oxygen position is not accurately described by a single Gaussian but as the sum of several Gaussians, one for non-displaced and two for each direction of the displaced oxygens. Furthermore, in contrast to the above displayed Gaussians the “average” oxygen position is always centred at the average position, i.e. 0 in the above case.

We agree that the discussion of this procedure was not outlined well enough. However, we feel that the main text is not the right place for this discussion. We dedicated a new Section 9 in the supporting information to this issue where we demonstrate that in the case of large static displacements as for YSZ we can directly quantify the shift using a one-dimensional chain as a model system.

To underline the argument that the variability in the shift magnitude is expected for the different probes, we fitted the shift magnitudes to our calculated model PDFs and arrive at a similar degree of variability with similar trends as observed in the experimental PDFs.

We hope that the reviewer agrees that these two points, now outlined in Section 9 of the supporting information and referenced on page 14 in the main text, indeed demonstrate that we can use the derived shift magnitudes to prove that the 3D PDF from electron diffraction can be used for quantitative measures.

The further suggestions for improvement are in the broader sense of linguistic or stylistic nature and I leave it to the authors to accept these or not.

- The term 'correlated disorder' is used several times. I find this somewhat unfortunate, since 'disorder' is more a global descriptor ('disorder model', 'disordered structure', etc., also the term 'disordered atom' is more associated with average structure properties) and less about very localised properties, which are seen in the 3D-DeltaPDF. I would prefer a continuous use of the term 'local order' as a more accurate description.

We agree and adapted the text to use local order throughout the text.

- In the second line of the abstract it is written about 'intentional introduction of disorder'. I think this is a bit unfortunate, since unintentional disorder can be just as interesting.

We agree and omitted the word intentional in the abstract.

- line 34ff: It would be good if some non-perovskite examples were also listed.

We added 2 additional references (refs. 6 and 9) of non-perovskite examples.

- line 158: new symbols $\Delta^{\text{sub}}_{\text{sup}}$ etc. are introduced without defining them. In the course of the text the meaning became clear to me, but without explanation they could be misunderstood.

We added a sentence that explains the subscript and superscript part of the symbols (page 12).

- Fig. 4: I cannot distinguish the colors for Zr and Y. Maybe a better color coding would be helpful.

We changed the colours in the figure and adapted the caption. The relevant interatomic vectors in the average structure are now better indicated.

- line 179: In the context of 3D-DeltaPDF features I would speak of 'densities' instead of 'intensities', because they are derived from scattering densities.

We adjusted the wording accordingly.

- lines 218 - 235: It is repeated several times that ED is more sensitive to light elements than X-ray diffraction, which affects the readability of this paragraph. Consider rewriting this paragraph.

We restructured the paragraph accordingly.

- Chapter 'Challenges ...' The challenge of multiple scattering in ED should be included in this paragraph. An additional paragraph discussing the possible influence of multiple scattering on the 3D-DeltaPDF would certainly be interesting, in particular, since some of the authors already have a lot of experience regarding average structure refinements based on 3D-ED diffraction. Would this experience be transferable to diffuse scattering?

We added multiple scattering and inelastic scattering as a challenge. In principle the formalism behind the multiple scattering is also applicable to large superstructures that are used to calculate diffuse scattering. For a more complete analysis of the effects of multiple scattering and also inelastic scattering we plan to utilize multislice simulations. This is work in progress, but due to the complexity of the problem and the computational efforts needed

for such simulations we would strongly prefer to leave this point to future research, where it can be discussed in the length and the thoroughness that we think is necessary and appropriate.

- Supplement chapter 4: I would write of 'dashed lines' instead of 'dotted lines' and again replace 'intensities' with 'densities'. Also one should write that the crossing points of the lines describe the average interatomic vector. In S4 there are four crossing points which should be listed in the caption.

We adapted the captions accordingly.

We would like to thank Reviewer #2 for their detailed insight and hope we could clarify the points raised here in the revised version.

Reviewer #3 (Remarks to the Author):

Dear Authors

The present study describes a successful use of electron diffraction data for conducting 3D local structure analysis on an inorganic material of technological interest, namely yttria stabilized zirconia. The general goal is to demonstrate that the analysis of local structure such as the spatial arrangement of defects and displacive disorder can be determined reliably and in 3D by electron diffraction, thus overcoming crystal size limitations imposed by single crystal X-ray and neutron diffraction. In this respect, this study presents a convincing use of electron diffraction data for conducting 3D-deltaPDF analysis, while conveniently comparing it with analogous results from X-ray and neutron 3D-deltaPDFs that have been previously published.

Overall, the study conveys an important message and highlights prominent challenges that need to be solved to make this technique more robust and widely used. As for the “quantitative” aspects of it, particularly standing out in the title and the manuscript itself, I believe that this study should put forward some more caveat and cautionary considerations. In the context of modern crystallography and materials science, this work constitutes an important step beyond the state of the art, which undoubtedly qualifies it for publication. I do have a few points, though, which in my view require attention and I will elaborate in the following. I therefore consider this work valuable as a Nature Communications article after some minor revisions are addressed.

1. Line 41. The Authors mention that the analysis of Bragg diffraction is well established. I would also add that is even “automated”, since less and less is left to humans in current crystallographic routines, and crystal structures can be solved by essentially autonomous software (such as the Autochem plugin by Rigaku) when data quality is not particularly problematic.

We included a statement related to automation in routine structure determination.

2. Line 68. The Authors state that the average interatomic vector of neighbouring atoms that vibrate in-phase “is smeared out by the average structure atomic displacement parameters (ADPs)”. I understand the intended meaning, but ADPs are used to model electron density distribution, therefore they cannot have any smearing effect themselves. I assume the Authors intended something like “is smeared out in the average structure, as modelled by atomic displacement parameters (ADPs)”. For this reason, I think this sentence should be rephrased to make it more accurate.

We rewrote the sentence accordingly.

3. Line 73. I would be more explicit on the crystal size, since “macroscopic” has different meanings depending on the context. Perhaps the Authors can consider using the word “micrometric”, and preferably pairing it with “currently” and “in general” to be on the safe side, since sub-microcrystal XRD can be performed in numerous synchrotrons worldwide.

We acknowledge the concern and adapted the sentence as suggested.

4. The Authors correctly point out that the importance of analyzing nanocrystals is primarily due to the fact that their correlated disorder might be different from their microcrystalline analogues. However, I find this aspect addressed very little when describing the interpretation of the data. For example, at line 130 the Authors describe the broadening of electron diffuse scattering data compared to the X-ray and neutron diffraction one, attributing it to the experimental setup simply based on the fact that both Bragg and diffuse scattering features are affected by broadening at the same time. I believe there are numerous microstructural characteristics that can differ between large microcrystals and nanocrystalline samples, affecting both diffuse scattering and the Bragg peaks’ profiles. This is not only a matter of short-ranged correlated disorder, but can also relate to coincidental change in mosaicity and crystallite size. Such effects can also be found in the PDF space, which would potentially invalidate the statement at line 146: “This is a direct consequence of the experimental broadening of the diffuse scattering”. I agree that broadening of diffuse scattering relates to correlation lengths in real space, but I am concerned with the use of “experimental” as in ‘due to the experimental setup’ (meaning in particular all the set of distortions that come with an electron diffraction experiment). I think this is an important aspect to clarify, and personally I would keep the option of having real structure effects impacting patterns and PDFs, which are therefore not due to experimental setup, while having analogous effects.

Addressing these aspects could also give the Authors a good chance to comment on the challenge of discriminating the two sources of broadening (structural changes vs. 'experimental' broadening), and the need of quantifying the instrumental sources of distortions in total scattering data from electron diffraction, as well as their effects in the PDF space. This aspect should also be kept in mind when, further in the manuscript, the Authors consider the quantitative differences of correlation lengths. In lines 202 and 205–207, these differences are attributed to the fact that probing matter with electrons and with X-rays or neutrons provides different lengths because of different sensitivity to “structural aspects”. However, I would also specify that differences due to the nanocrystalline nature of the sample and its different, unknown microstructure can also be a cause of such varying correlation lengths.

We agree that structural origins due to microstructure variations cannot be excluded and indicated so in the revised version of the main text. However, we are convinced that the setup related broadening is the main cause of the reduced observed correlation length. To elucidate this point, we included a section (section 4 in updated SI) in the supporting information that estimates an instrumental resolution function based on the FWHM of an observed Bragg reflection in the reconstruction and henceforth calculate the structural correlation length from the width of the observed diffuse scattering.

We attribute the main cause of this observed broadening to inelastically scattered electrons that are responsible for the halo around the primary beam in reciprocal space but at the same time also broaden all features in the diffraction pattern to the

same degree. Unfortunately, we currently don't have a setup available with an energy discriminating detector that can still yield the desired reciprocal space coverage to fully quantify this analysis.

5. The proposed PDF analysis is useful and in line with what previously published based on X-ray and neutron data, but there is an aspect that I did not find addressed explicitly: the impact of vacancy defects on the PDF space. The interpretation appear focused on the consequences of vacancies on structural distortions, leaving the reader with the impression that the only information available from the 3D-deltaPDFs is displacive disorder, and no localisation and quantification of correlation signals due to lack of density (presence of defects) can be distinguished. This might indeed be true, and I believe that, in case, the Authors should state this clearly to better guide the readers. It would be useful to mention qualitatively and briefly what kind of signals can be expected when vacancy defects are present and why they are not interpretable from the available experimental delta-PDFs of nanocrystalline yttria stabilized zirconia.

We agree and we added further statements in the results section that describe that in this sample material the displacive local order is dominating here (page 10), commenting on the limited detectability of chemical short-range order in our sample material (page 6, page 10, page 13).

6. The Authors address the quantitative reliability of the 3D-deltaPDF analysis by comparing correlation lengths in terms of positions of maxima and minima between X-ray, neutron and electron PDFs. I find this a relatively weak part that could be improved, and I will elaborate in the following.
In my view, there are two aspects that can be considered concerning 'quantitative reliability' in the PDF space: the positions of the correlation vectors, and the numerical values (maxima/minima or integrals) of the correlation signals. It appeared to me that the Authors focus selectively on the first aspect. Nonetheless, the second, too, contains important defect structure information, which can be made less reliable due to artefacts from data processing, instrumental broadening, or multiple elastic and inelastic scattering. It is not possible to use comparisons with XPDFs or with neutron PDFs for assessing quantitative reliability, because both shift magnitudes and integrals at specific correlation vectors can be influenced by the fact that nanocrystals can have different local structures, and because, as the Authors pointed out, "different radiations are sensitive to different structural aspects". For this reason, I believe that using such comparisons to evaluate the quantitative reliability does not help much, and indeed this section concludes with the Authors acknowledging that differences are "to be expected".

The integrals of the 3D- Δ PDFs do indeed give valuable insight on the chemical short-range order and we added a paragraph in the introduction that explains that in the sample material YSZ displacement disorder is the dominating effect.

To underline that the variations of the determined shift magnitudes can indeed be explained by the different sensitivities of the different radiation types we included a new section 9 in the supporting information: Here, we use our simplistic model crystal and determine the shift magnitudes from the resulting 3D- Δ PDFs calculated from electron, x-ray and neutron diffraction. As the underlying model crystals are identical, Bragg peaks in computational data are exactly one voxel and there are no other sources of background, this comparison enables a quantification of differences in shift magnitudes due to the different probes. We find that the variations we observe here are similar to our experimental observations, where shifts determined from x-ray

and electron diffraction experiments are within the uncertainty of each other while due to the larger difference in scattering length the shift as determined from neutron diffraction experiments show larger deviations. The trend of the deviation of the neutron shift is the same in the computational model analysis as observed in the experimental data and so we believe that our quantitative analysis indeed gives reliable results.

I consider these comparisons nonetheless useful, but I wonder whether the Authors could be more transparent with respect to this ambiguity in quantitative reliability, and formulate how it could be determined more appropriately. This remains an open question that does not stand out enough, in my opinion. It could also become an important point to be presented in the “challenges that need addressing” section: how can we validate our 3D-deltaPDF from electron diffraction data, since we cannot compare it with neutron and X-ray PDF analysis due to the different nature of the beam-sample interaction and the possibly different local structure?

We agree that the validation of the derived local order model is a challenge and added an additional paragraph in the discussion section that comments on this issue (page 19).

7. The supplementary material is missing the pristine reciprocal space reconstructions of the main planes of the sample ($hk0$, $h0l$, $0kl$). Since the production of diffuse scattering reconstructions required not only a reasonable symmetry averaging, but also other steps such as “Karen” filtering, punch-and-fill, and gaussian-dampening, it would be most informative for the readers—especially those who are willing to conduct similar analyses in the future—to have an idea of the pristine reconstruction and possibly some intermediate steps of the data processing. Furthermore, punch-and-fill procedures have also an important effect on the features observed in the 3D-deltaPDF, especially the choice of punch radius. Indeed, different punch radii may change the integrals and shift of correlation peaks, as the Patterson function might partly contribute, which is a likely possibility when the diffuse scattering is primarily originating from correlated distortions in the structure. Ideally, the chosen punch radius should be justified, and the effect of different punch radii could be briefly shown to give an idea of how much this process affects the features that are discussed in the manuscript. This would also provide useful insights on the quantitative side of this analysis, since it is a central aspect the Authors aimed at.

We added the unprocessed data sections and a figure showing the effect of the punch radius in the supporting information for reference (Fig. S6).

8. When addressing the challenges of the method, the Authors describe three main points: limited reciprocal space coverage, sample stability to the electron beam, and dynamic range of the detector. However, I think an important aspect is missing: the presence of non-kinematical elastic or inelastic scattering. Kikuchi diffraction and other inelastic scattering events can produce broad band-like intensities that might be problematic to distinguish and separate from the data, adding undesired features to the 3D-deltaPDFs. This is particularly relevant for inorganic samples such as the one addressed in this study, where the Authors took care of preparing it as a thin lamella. The strong influence of sample thickness, ubiquitous presence of undesired multiple and inelastic scattering, and the possibility of using energy-filtered diffraction to at least eliminate most of these intensities, would all be important aspects to be discussed in this last section.

Furthermore, the challenge of validating the electron 3D-deltaPDF analysis with

complementary techniques for the reasons described above can also be proposed as an open challenge, since there is no standard practice, as far as I am aware, for this purpose.

We included multiple and inelastic scattering as an open challenge and suggest a detailed quantitative investigation of the effects for the future. e.g., using multislice simulations.

9. I did not find, in the cited literature, the use of electron diffraction data for 1D PDF analysis, as reported by Gorelik et al. in 2015 (Microscopy and microanalysis, 2015, 21(2), 459–471) and later improved towards a quantitatively reliable technique (Gorelik et al. Acta Cryst. 2019, B75, 532–549). This is not only highly relevant to this study, but it might also allow a better quantitative reliability assessment of the electron 3D-deltaPDF analysis by comparing the electron 3D-deltaPDF with its 1D analogue, which, in turn, can be more reliably compared to 1D XPDF.

We added the references 15 and 16 when discussing 1D PDF (page 3).

10. Just a couple of writing slips I noticed. Line 44: Since the subject is “powder pair distribution function analysis”, I believe the subsequent verb should be “has been”. In the caption of Figure 4, line 175 reads “diffraction patterns a simplified model”. I imagine “from” should be inserted before “a simplified model”.

We corrected the mentioned writing slips.

Finally, I would like to congratulate the Authors on this beautiful work and wish them all the best for the revision process.

We tank Reviewer #3 for their insightful comments and hope we could address the concerns in the revised version.

REVIEWERS' COMMENTS

Reviewer #1 (Remarks to the Author):

The authors have addressed all the issues I can find in the paper, and I recommend the manuscript be accepted for publication.

One very minor point: Readers might find a statement in the text about the relationship of these derived values to the displacements in the crystal helpful (as in Ref. 23, where the neutron value is taken as the best estimate for the oxygen-vacancy). That said, such a statement might overly simplify the qualitative tendencies mentioned in this text, so I am not sure it needs to be included, and I leave this as a suggestion to the authors and editors.

Reviewer #2 (Remarks to the Author):

I have now looked through the corrected manuscript carefully. My comments, and as far as I have seen also those of the other referees, have been satisfactorily taken into account. The paper, which was already very nice, has now gained further in quality and I can recommend it for publication without reservation. I'm looking forward to seeing the final paper 'printed'.

Reviewer #3 (Remarks to the Author):

Dear Authors

I am glad to confirm that every comment from my side has been adequately addressed and the missing information added to the manuscript/SI.

One last detail I unfortunately missed during my first review: ADPs should be "anisotropic displacement parameters", not to be confused with the isotropic Uiso.

For reference: <https://www.iucr.org/resources/commissions/crystallographic-nomenclature/adp>

I kindly encourage the authors to correct the text to "anisotropic displ. param. (ADP)" or keep the word "atomic" while avoiding mentioning the acronym ADP, since it is not used anywhere else in the manuscript so it is not strictly needed.

Best wishes and my sincere congratulations for this nice work

RESPONSE TO REVIEWERS' COMMENTS

Reviewer #1 (Remarks to the Author):

The authors have addressed all the issues I can find in the paper, and I recommend the manuscript be accepted for publication.

One very minor point: Readers might find a statement in the text about the relationship of these derived values to the displacements in the crystal helpful (as in Ref. 23, where the neutron value is taken as the best estimate for the oxygen-vacancy). That said, such a statement might overly simplify the qualitative tendencies mentioned in this text, so I am not sure it needs to be included, and I leave this as a suggestion to the authors and editors.

We thank Reviewer #1 for their kind words and helpful comments for the revised version. As suggested, we inserted a sentence on page 11 to reference that the neutron value is the most reliable value.

Reviewer #2 (Remarks to the Author):

I have now looked through the corrected manuscript carefully. My comments, and as far as I have seen also those of the other referees, have been satisfactorily taken into account. The paper, which was already very nice, has now gained further in quality and I can recommend it for publication without reservation. I'm looking forward to seeing the final paper 'printed'.

We thank Reviewer #2 for their kind words and helpful comments for the revised version.

Reviewer #3 (Remarks to the Author):

Dear Authors

I am glad to confirm that every comment from my side has been adequately addressed and the missing information added to the manuscript/SI.

One last detail I unfortunately missed during my first review: ADPs should be "anisotropic displacement parameters", not to be confused with the isotropic Uiso.

For reference: <https://www.iucr.org/resources/commissions/crystallographic-nomenclature/adp>

I kindly encourage the authors to correct the text to "anisotropic displ. param. (ADP)" or keep the word "atomic" while avoiding mentioning the acronym ADP, since it is not used anywhere else in the manuscript so it is not strictly needed.

Best wishes and my sincere congratulations for this nice work

We thank Reviewer #3 for their kind words and apologize for misusing the acronym ADP. As suggested, we deleted the acronym in brackets.